# TABASCO: A Fast, Simplified Model for Molecular Generation with Improved Physical Quality

**Carlos Vonessen**[*][†]                                                                 *cvonessen@ethz.ch*
*ETH Zurich*

**Charles Harris**[*]                                                                      *cch57@cam.ac.uk*
**Miruna Cretu**[*]                                                                        *mtc49@cam.ac.uk*
**Pietro Liò**                                                                             *pl219@cam.ac.uk*
*University of Cambridge*

**Reviewed on OpenReview:** *https://openreview.net/forum?id=Kg6CSrbXl4*

## Abstract

State-of-the-art models for 3D molecular generation are based on significant inductive biases: $SE(3)$ equivariance, permutation invariance and graph message-passing networks to capture local chemistry, yet the generated molecules struggle with physical plausibility. We introduce TABASCO which relaxes these assumptions: The model has a standard non-equivariant transformer architecture, treats atoms in a molecule as sequences and does not explicitly model bonds. The absence of equivariant layers and message passing allows us to simplify the model architecture and scale data throughput. On the GEOM-Drugs and QM9 benchmarks TABASCO achieves state-of-the-art POSEBUSTERS validity and delivers inference roughly $10\times$ faster than the strongest baseline, while exhibiting emergent rotational equivariance without hard-coded symmetry. Our work offers a blueprint for training minimalist, high-throughput, unconditional generative models and the resulting architecture is readily extensible to future conditional tasks. We provide a link to our implementation at https://github.com/carlosinator/tabasco.

## 1 Introduction

In recent years, there has been growing interest in using diffusion models as generative methods for molecular design (Du et al., 2024; Schneuing et al., 2022; Hoogeboom et al., 2022; Vignac et al., 2023; Dunn & Koes, 2024; Irwin et al., 2025). Much of the literature converges on design principles believed to be essential for high-quality molecular generation. First, models are typically SE(3)-equivariant, a symmetry prior that serves as a strong inductive bias (Hoogeboom et al., 2022). Second, message-passing graph neural networks (GNNs) are widely used to capture many-hop, context-dependent interactions between atoms (Hoogeboom et al., 2022; Schneuing et al., 2022; Irwin et al., 2025; Dunn & Koes, 2024; Schneuing et al.), making them permutation-invariant. Third, recent work emphasises flow-matching objectives that rely

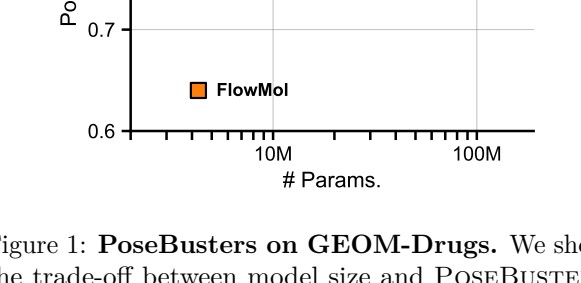

Figure 1: **PoseBusters on GEOM-Drugs.** We show the trade-off between model size and POSEBUSTERS performance for TABASCO and baselines.

---

[*]Core contributor.
[†]Work done as a visitor at the University of Cambridge.

on coupled optimal transport (OT) (Tong et al., 2023) or incorporate heavily structured, domain-informed priors (Dunn & Koes, 2024; Irwin et al., 2025). Despite incorporating these inductive biases, current models continue to struggle with physical plausibility, often failing to produce chemically coherent structures or accurately recover fundamental features of protein–ligand binding (Buttenschoen et al., 2024; Harris et al., 2023).

In parallel, a growing body of work explores scaling up simpler model architectures, most notably Transformers (Vaswani et al., 2017), across adjacent domains. A prominent example is AlphaFold3 (Abramson et al., 2024), which achieves strong performance on physical plausibility benchmarks (Buttenschoen et al., 2024) despite omitting many of the conventional inductive biases, including equivariance. Similarly, recent generative models for protein backbone design have demonstrated competitive results with minimal architectural complexity, provided they are scaled appropriately (Geffner et al., 2025). Simplification of model architecture and removal of inductive biases for conformer generation has also proven successful (Wang et al., 2024). In parallel to this work, (Joshi et al., 2025) explore using non-equivariant latent diffusion for generating small molecules.

In this work, we aim to distill the core components of diffusion-based molecular generation and ask: *Which concepts are necessary to build high-performing models?* We introduce TABASCO (Transformer-based Atomistic Bondless Scalable Conformer Output), a scalable model that achieves state-of-the-art performance on unconditional molecular generation benchmarks. Our contributions are as follows:

(i) **State-of-the-art physical quality on GEOM-Drugs and QM9.** TABASCO surpasses previous models in POSEBUSTERS validity, and achieves a $10\times$ speed-up during sampling (Sections 4.3 and 4.4).

(ii) **Greatly streamlined approach.** Our model omits both explicit bond modelling, equivariant layers and permutational invariance, and instead utilizes a standard Transformer, sinusoid encodings, and random rotations of data points (Section 3).

(iii) **Study of model behaviour without explicit symmetry.** Under this new approach, we study the behaviour of the model with respect to permutational invariance and SE(3) equivariance.

    (a) We find evidence that breaking permutational invariance with positional encodings significantly improves performance by helping distinguish atoms during early denoising (Section 4.5).

    (b) We find that completely removing random rotations and SE(3) equivariance constraints has a negligible impact on performance (Section 4.6).

(iv) **Physically-constrained last-mile guidance.** We introduce an separate distance-bounds guidance algorithm that further improves POSEBUSTERS validity without requiring force-field-based relaxation or additional parameters (Section 5).

## 2 Background and Related Work

### 2.1 Flow-Matching Models

Flow-matching (FM) is a generative modelling framework that learns to transport samples from a source distribution (e.g., noise) to a target distribution (e.g., data) by directly estimating the time-dependent velocity field of a probability flow (Lipman et al., 2023; Albergo & Vanden-Eijnden, 2022).

Given a pair of samples $(\mathbf{x}_0, \mathbf{x}_1)$ from source and target distributions, one defines a continuous interpolation $\mathbf{x}_t = (1-t)\mathbf{x}_0 + t\mathbf{x}_1$, and a target velocity $u_t = \mathbf{x}_1 - \mathbf{x}_0$. A neural field $v_\theta(\mathbf{x}_t, t)$ is then trained to match this velocity using the mean squared error:

$$\mathcal{L}_{\text{FM}} = \mathbb{E}_{t,(\mathbf{x}_0,\mathbf{x}_1)} \left[ ||v_\theta(\mathbf{x}_t, t) - u_t||_2^2 \right]. \tag{1}$$

Flow-matching enables efficient generation via deterministic integration (e.g., using an ODE solver), and has been shown to improve sampling speed and stability over score-based diffusion models (Dunn & Koes, 2024; Irwin et al., 2025).

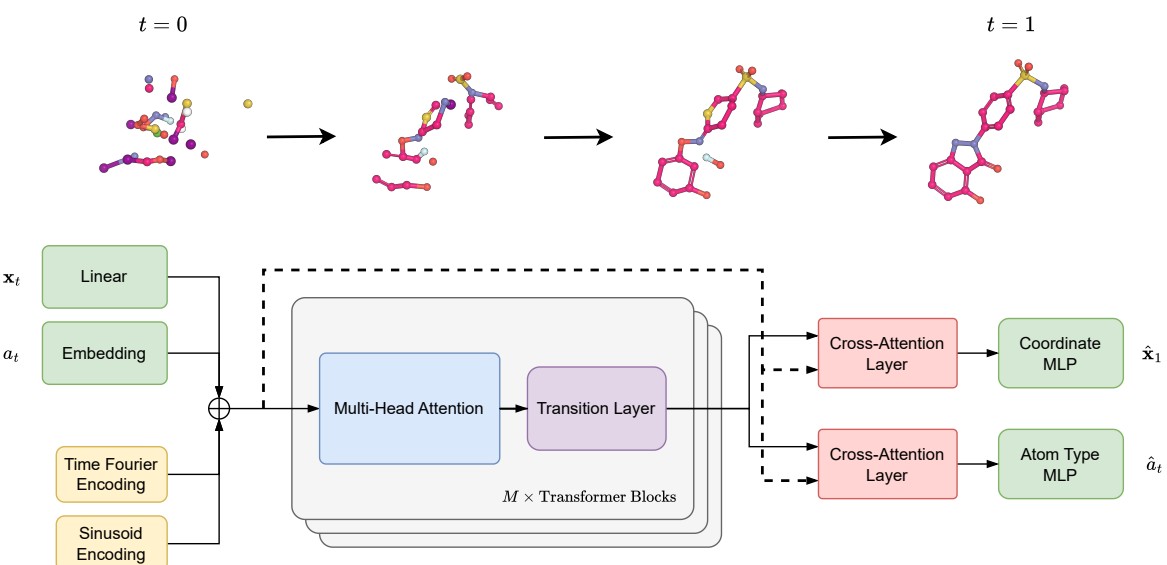

Figure 2: Top: **Interpolation between noise and data.** Bottom: **TABASCO model architecture.** $x$ and $a$ denote coordinates and atom types, respectively.

## 2.2 Generative Models for 3D Molecule Design

Early works used standard continuous diffusion processes on coordinate and atomic features, where bond connectivity was determined by chemoinformatics software (Hoogeboom et al., 2022; Schneuing et al., 2022). This process often resulted in low-quality conformers that were not fully-connected or violated atomic valences. MiDi (Vignac et al., 2023) improved on this by applying discrete diffusion to both the atom types as well as generated a full bond matrix end-to-end, which significantly increased stabilty and bond connectivity. EQGAT-diff (Le et al., 2023) explored the design space of equivariant diffusion models, creating a custom attention-based equivariant architecture to allow for interaction between continuous and discrete features. Further work introduced more advanced model architectures (Morehead & Cheng, 2024; Hua et al., 2024), additional losses (Xu et al., 2024), alternative transport strategies (Song et al., 2023), and geometric latent diffusion (Xu et al., 2023; Joshi et al., 2025). FlowMol (Dunn & Koes, 2024) and SemlaFlow (Irwin et al., 2025) use flow-matching for generation of coordinates, atom types and bonds. Both methods proposed new architectures and showed great improvements in speed versus diffusion based approaches.

## 3 TABASCO: Fast, Simple, and High-Quality Molecule Generation

**Overview and Motivation**   Our goal in this work is to identify the simplest possible model architecture that can generate physically realistic small molecules at scale. Our motivation stems from the observation that recent progress in protein structure generation has demonstrated the surprising power of non-equivariant Transformer architectures when scaled appropriately (Abramson et al., 2024; Geffner et al., 2025; Wang et al., 2024). Based on these results, we began our experiments with a deliberately stripped-down, non-equivariant Transformer backbone for molecular generation. We also chose to exclude explicit bond information from the model. While many existing models treat bonds as a distinct modality (Irwin et al., 2025; Dunn & Koes, 2024; Vignac et al., 2023), in line with earlier work (Schneuing et al., 2022) we hypothesised that if coordinate generation is sufficiently accurate, bond information can be successfully imputed with standard chemoinformatics tools. This perspective allowed us to further simplify the architecture while focusing on improving conformer quality. Physical realism, as measured by POSEBUSTERS validity, was the primary metric guiding design decisions. Modules and heuristics in the approach that did not contribute to this metric were pruned, resulting in a lean, fast, and extensible model that maintains strong performance without relying on specialised architectural components.

### 3.1 Model Architecture

In contrast to most prior work in unconditional molecular generation, we adopt a simplified non-equivariant Transformer architecture (see Figure 2) without self-conditioning. Atom coordinates and types are jointly embedded along with time and sequence encodings and are passed through a stack of Transformer blocks. We add a single cross-attention layer for each domain and process these outputs in MLP heads for atom types and coordinates. We report resulting performance and model ablations in Section 4.3.

### 3.2 Training Objective

We optimize coordinates with Euclidean conditional flow-matching (CFM) (Tong et al., 2023; Albergo & Vanden-Eijnden, 2022) and atom types with discrete CFM which is parametrized based on the Discrete Flow Models (DFM) framework (Campbell et al., 2024). Concretely, consider a molecule with $N$ atoms, ground-truth coordinates $\mathbf{x_1}$ and atom types $a_1$. Coordinates are partially noised with $\mathbf{x_t} = t \cdot \mathbf{x_1} + (1-t) \cdot \epsilon$, where the noise is distributed with $\epsilon \sim \mathcal{N}(0, I)$. Noisy atom types $a_t$ are obtained by interpolating between atom type probabilities and sampling from a categorical distribution $a_t \sim \mathrm{Cat}\left(t \cdot \delta(a_1) + (1-t) \cdot \frac{1}{N}\right)$, where $\delta(\cdot)$ creates a one-hot encoding (Campbell et al., 2024). During training the model takes $\mathbf{x_t}$ and $a_t$ and learns to predict the endpoint of the trajectory. The continuous coordinate objective becomes

$$L_{\mathrm{metric}}(\mathbf{x}) = \mathbb{E}_{\epsilon,t}\left[\frac{1}{N}||\hat{\mathbf{x}}_1^\theta(\mathbf{x}_t, t) - \mathbf{x}_1||_2^2\right]. \tag{2}$$

The discrete atom type objective is the cross-entropy loss

$$L_{\mathrm{discrete}}(a) = \mathbb{E}_t\left[-\sum_i a_i \log(\hat{a}_1(a_t, t))\right]. \tag{3}$$

We combine these into a multi-objective formulation with weighing factor $\lambda_{\mathrm{discrete}} \in (0, 1]$, as

$$L_{\mathrm{total}}(\mathbf{x}, a) = L_{\mathrm{metric}}(\mathbf{x}) + \lambda_{\mathrm{discrete}} \cdot L_{\mathrm{discrete}}(a). \tag{4}$$

During training we sample from $t \sim \mathrm{Beta}(\alpha, 1)$, where $\alpha$ is a hyperparameter we ablate in Appendix B. As $t \to 1$ the model's behaviour approaches the identity function, due to the chosen endpoint formulation. Inspired by Le et al. (2023); Salimans & Ho (2022) and to ensure the model can still learn precise atom placement even as losses approach zero as $t \to 1$, we weigh the loss with $\beta(t) \cdot L_{\mathrm{total}}(\mathbf{x}_t, a_t)$ based on the sampled time $t$, with

$$\beta(t) = \min\left\{100, \frac{1}{(1-t)^2}\right\}. \tag{5}$$

### 3.3 Sampling

We generate molecules with TABASCO by simulating a system of coupled stochastic differential equations:

$$\mathrm{d}\mathbf{x}_t = \mathbf{v}_t^\theta(\mathbf{x}_t, a_t)\mathrm{d}t + g(t)\,\mathbf{s}_t^\theta(\mathbf{x}_t, a_t)\mathrm{d}t + \sqrt{2g(t)\gamma}\,\mathrm{d}W_t,$$
$$\partial p_t = R_t(\mathbf{x}_t, a_t)^\top p_t \tag{6}$$

where $p_t$ describes the probability of each atom type at time $t$. We estimate the velocity with $\mathbf{v}_t = \frac{\mathbf{x}_1 - \mathbf{x}_t}{1-t}$ from the models endpoint prediction $\hat{\mathbf{x}}_1$ at time $t$, and the score with $\mathbf{s}_t = \frac{t\mathbf{v}_t - \mathbf{x}_t}{1-t}$. We refer the reader to Geffner et al. (2025) and Campbell et al. (2024) on which we base our coordinate and atom type sampling strategies, for more in-detail discussions of this approach. To improve sample quality, we apply a logarithmic discretization scheme on $t \in [0, 1]$ with more fine-grained steps near the end of denoising. We also scale the score $\mathbf{s_t}$ and the Gaussian noise component $\mathrm{d}W_t$ by $g(t)$, setting it to zero as $t \to 1$ (see Appendix B).

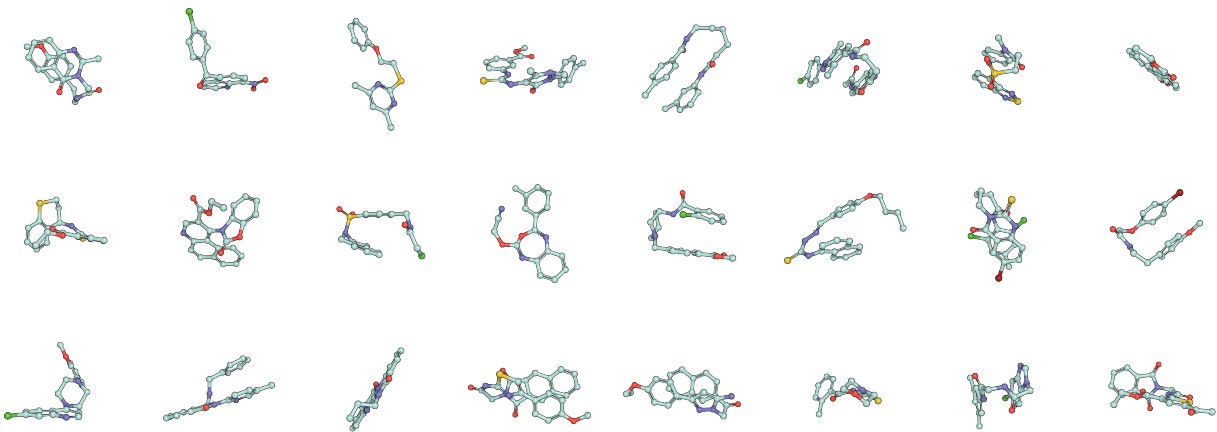

Figure 3: **Sampled molecules from TABASCO**.

### 3.4 Ordering Atoms as Sequences

Transformers operate in a bag-of-tokens fashion unless provided with additional information about the absolute or relative positions of those tokens. Unlike text or protein sequences, small molecules lack a natural linear ordering that reflects their 3D structure. While formats such as SMILES and InChI offer consistent ways to linearise molecular graphs, the ordering in these representations does not strictly correspond to spatial proximity. However, the SMILES ordering is deterministically derived, typically via a depth-first traversal starting from a canonical root atom (Weininger et al., 1989), which still imparts some semantic structure onto the linearized sequence. In practice, many neighbouring atoms in the SMILES string are also spatially or chemically proximate in the molecule. Accordingly, we include sinusoidal positional encodings based on the atom indices in the SMILES sequence. We hypothesise that this implicit locality helps the model distinguish atoms during denoising when atom coordinates contain limited information and thus create molecules with higher physical validity. We ablate the effect of the positional encodings in Section 4.5.

## 4 Experiments

### 4.1 Training Setup

We train TABASCO on GEOM-Drugs (Axelrod & Gomez-Bombarelli, 2022), a dataset of 1M high-quality conformers of drug-like molecules. We use the splits from Vignac et al. (2023) and, following Irwin et al. (2025), we discard molecules with more than 72 heavy atoms from the training dataset, accounting for 1% of the data. During testing, we sample the number of atoms from the distribution of molecule sizes in the test set, which was left unchanged. We separately train and evaluate our model on QM9 (Ramakrishnan et al., 2014), which is an enumeration of all physically plausible molecules with up to nine atoms. Since the molecules in QM9 are small in size and GEOM-Drugs is considered to be a significantly more challenging dataset, most results are reported for models trained on GEOM-Drugs, except for Table 3.

We train three TABASCO models at three sizes: Both TABASCO-mild (3.7M params.) and TABASCO-hot (15M params.) were trained on two 80GB A100 GPUs for 36 hours at a learning rate of 0.001. TABASCO-spicy (59M params.) was trained on the same resources for 72 hours with a learning rate of 0.0005 (see Appendix D). During training, we augment each batch with 8 random rotations of the same molecules to improve equivariance. We apply an Exponential Moving Averaging (EMA) with decay strength 0.999 to the model weights, which we ablate in Section B. We compare our models against EQGAT-diff (Le et al., 2023), FlowMol (Dunn & Koes, 2024), SemlaFlow (Irwin et al., 2025), and ADiT (Joshi et al., 2025) (see Appendix A).

Table 1: **Physical validity results for all models.** A molecule is only POSEBUSTERS-valid overall if it passes all subtests. Higher is better on all metrics. [α]Due to computational constraints, we evaluate statistics on GEOM-Drugs on a random subset of 20K training molecules.

| Model | Overall | Connected | Bond Lengths | Bond Angles | No Clashes | Rings Flat | Double Bonds Flat | Internal Energy |
|---|---|---|---|---|---|---|---|---|
| GEOM-Drugs[α] | 0.94 | 1.0 | 1.0 | 1.0 | 0.94 | 1.0 | 1.0 | 1.0 |
| EQGAT-diff | 0.84 | 0.91 | 0.94 | 0.94 | 0.91 | 0.94 | 0.93 | 0.90 |
| FlowMol | 0.64 | 0.68 | 0.81 | 0.81 | 0.78 | 0.81 | 0.80 | 0.80 |
| SemlaFlow | 0.88 | 0.91 | 0.94 | 0.94 | 0.92 | 0.94 | 0.94 | 0.94 |
| ADiT | 0.87 | 0.96 | 0.95 | 0.95 | **0.94** | **0.98** | **0.98** | 0.95 |
| TABASCO-mild | 0.85 | 0.96 | 0.94 | 0.94 | 0.87 | 0.96 | 0.96 | 0.94 |
| TABASCO-hot | 0.91 | **0.98** | **0.98** | **0.98** | 0.92 | **0.98** | **0.98** | **0.97** |
| TABASCO-spicy | **0.92** | 0.97 | 0.97 | 0.97 | 0.92 | 0.97 | 0.97 | **0.97** |

## 4.2 Evaluation Metrics

We employ POSEBUSTERS as our main metric for measuring conformer quality, as its array of tests are designed to test for physical plausibility (Buttenschoen et al., 2024). A molecule is POSEBUSTERS-valid only if it passes an array of tests, including: all atoms are connected, valid bond lengths, valid bond angles, no steric clashes, flat aromatic rings, flat double bonds, and the internal energy is comparable to reference conformers. This stricter evaluation is necessary, since in existing generative models for 3D molecule generation, most other metrics have been saturated (Irwin et al., 2025).

We further evaluate generated molecules on several well established metrics: (i) **Validity**: Whether a molecule can be sanitized with RDKIT, (ii) **Novelty**: Whether the canonical SMILES of the molecule is not present in the training set, (iii) **Diversity**: Tanimoto similarity of molecule fingerprints, (iv) **Strain Energy** (Harris et al., 2023): Energy of the molecule compared to low energy conformers, (v) **Root Mean Square Deviation** (RMSD): The averaged distance between the atoms of two molecules when comparing molecules in our guidance experiment, (vi) **Jensen-Shannon Divergence** (JSD): We extract several features from generated molecules and reference molecules and test how much the features distribution deviates from the reference distribution. The extracted features are: bond lengths, bond angles, dihedral angles, frequency of a given bond pair or triplet, average bonds per atom type, number of rings per molecule, and bond type (see Appendix C).

## 4.3 TABASCO Results Across Metrics

**Results on Physical Validity.** Our physical validity results are shown in Table 1, and generated example molecules are shown in Figure 3. TABASCO-spicy (59M) and TABASCO-hot (15M), surpass all prior methods in physical plausibility, raising the POSEBUSTERS validity from the previous state-of-the-art of 0.88 to 0.92 (see Figure 1). Interestingly, most of the gain is achieved by the 15M parameter TABASCO-hot variant, with only modest improvements from further scaling to 59M, suggesting diminishing returns beyond this point. This may be explained by the fact that in GEOM-Drugs only 94% of molecules are POSEBUSTERS-valid. Alternatively, we also highlight that increasing returns to scale might become more apparent with an larger training budget. Earlier models such as FlowMol, which perform well on traditional metrics (see Table 2), show significantly lower physical validity (0.64), further highlighting the need for domain-aware evaluation such as POSEBUSTERS.

The tests contained in POSEBUSTERS also reveal in which ways TABASCO improves physical validity of generated molecules compared to previous methods: Notably, in contrast to most previous methods, TABASCO does not explicitly model bonds, but improves connectivity, bond length and bond angle validity. This indicates that sufficiently precise coordinate modelling can suffice to accurately impute bonds post-hoc. We also observe that ADiT and TABASCO, the only non-equivariant models, achieve significantly higher validity scores for steric clashes, flat aromatic rings, and flat double bonds compared to the equivariant models.

**Results on Traditional Metrics.** Table 2 shows a summary of results on traditional generation metrics. We observe that all variants of TABASCO maintain strong molecular diversity and novelty ($\sim 0.89$), indicating

Table 2: **Results on established metrics for GEOM-Drugs.** We generate 3K molecules for each method. $^\alpha$Due to computational constraints, we evaluate statistics on GEOM-Drugs on a random subset of 20K training molecules.

| Method | # Params. | Validity↑ | Novelty↑ | Diversity↑ | Strain Energy↓ | Time↓ (s) |
|---|---|---|---|---|---|---|
| GEOM-Drugs$^\alpha$ | - | 1.0 | 0.0 | 0.90 | - | - |
| EQGAT-diff | 12M | 0.94 | 0.94 | 0.90 | 360.19 | 4310.94 |
| FlowMol | 4.3M | 0.81 | 0.81 | **0.91** | 34.20 | 362.22 |
| SemlaFlow | 22M | 0.93 | 0.93 | **0.91** | 18.20 | 201.22 |
| ADiT | 150M | **0.98** | **0.97** | **0.91** | 46.36 | 521.21 |
| TABASCO-mild | 3.7M | 0.95 | 0.93 | 0.89 | 21.32 | **5.9** |
| TABASCO-hot | 15M | **0.98** | 0.93 | 0.88 | **14.16** | 10.67 |
| TABASCO-spicy | 59M | 0.97 | 0.90 | 0.89 | 15.07 | 19.77 |
| TABASCO-spicy w/ guidance | 59M | 0.97 | 0.92 | 0.89 | 19.23 | 131.80 |

that performance improvements do not compromise sampling breadth and generalisation. We note that, inline with previous works, for all evaluated methods Novelty and Diversity are computed over all molecules, including molecules that do not pass physical validity, which may lead to invalid molecules distorting the scores in both methods. Table 2 also shows that that the time required to sample from TABASCO is at least 10× lower than for any previous method, and up to 100× faster than some prior baselines, offering a practical advantage for large-scale or iterative workflows. In Appendix C we additionally compare the JSD of several features of generated molecules and GEOM-Drugs molecules for TABASCO-hot (15M) and three baselines.

## 4.4 Performance Comparison on QM9

We further train and evaluate our model on the benchmark dataset QM9 (Ramakrishnan et al., 2014) and compare the performance to previous methods in Table 3. We observe that TABASCO achieves state-of-the-art POSEBUSTERS scores, although showing low novelty scores on QM9.

Table 3: **Results on the QM9 Dataset.**

| Method | # Params. | Validity↑ | Novelty↑ | Diversity↑ | PoseBusters↑ | Strain Energy↓ |
|---|---|---|---|---|---|---|
| EQGAT-diff | 12M | 0.99 | **0.99** | 0.89 | 0.94 | 9.10 |
| FlowMol | 4.3M | 0.97 | 0.97 | 0.92 | 0.92 | 17.81 |
| SemlaFlow | 22M | 0.99 | **0.99** | 0.89 | 0.95 | 4.69 |
| TABASCO-mild | 3.7M | 0.98 | 0.31 | 0.91 | **0.98** | **2.31** |
| TABASCO-hot | 15M | 0.99 | 0.32 | 0.92 | **0.99** | 3.32 |
| *w/o pos. encodings* | 15M | **1.00** | 0.34 | **0.93** | 0.93 | 17.10 |

## 4.5 Breaking Permutational Invariance

In Figure 4 (right) we observe that introducing sequence positional encodings yields higher quality molecules compared to treating atoms in a bag-of-words fashion. In Table 3 we observe that this performance gap is much smaller for models trained on QM9 compared to TABASCO on GEOM-Drugs (see Figure 4, right). A possible explanation for this is that the much smaller number of atoms per molecule compared to GEOM-Drugs makes it easier to distinguish atoms and place them with respect to each other even without positional encodings. Additionally, in Figure 5 we show examples of failure modes we observed repeatedly in models without positional encodings.

**Disentangling the Effect of Positional Encodings** We hypothesize that the difference in generative quality when adding positional encodings may stem from early steps in molecule denoising, when the atomic coordinates are very noisy and positional encodings can provide a signal about the relative positions of

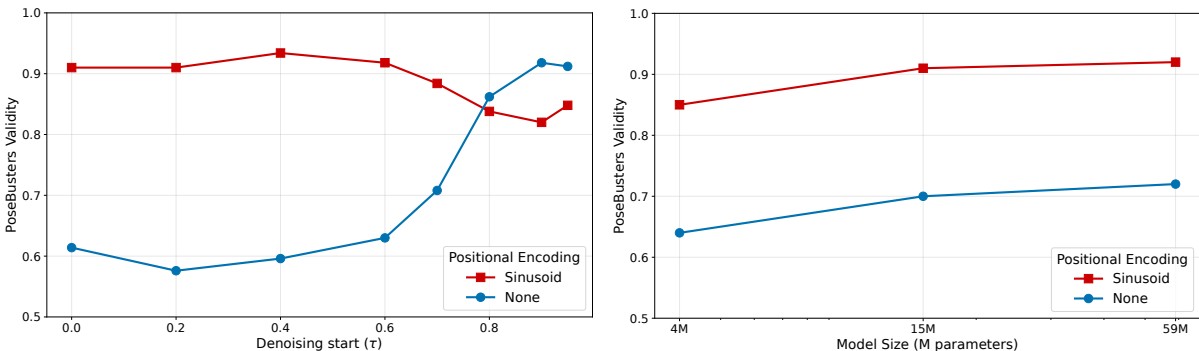

Figure 4: Left: 15M parameter model with/without positional encodings, POSEBUSTERS when starting denoising from different noise levels on test molecules from GEOM-Drugs. Right: Model performance across parameter scales with/without positional encodings when trained on GEOM-Drugs.

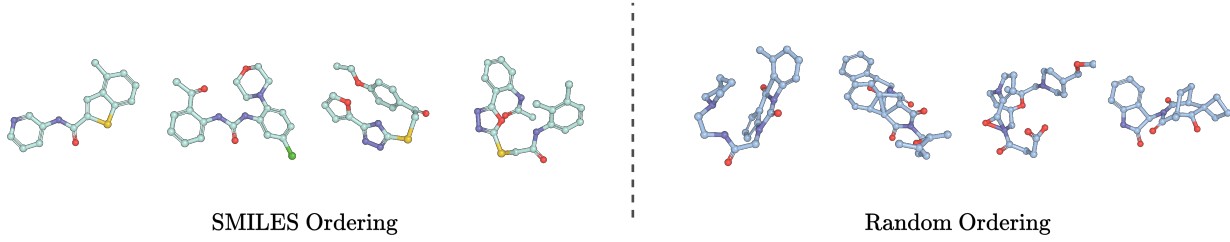

SMILES Ordering                                          Random Ordering

Figure 5: **Importance of atom ordering in TABASCO.** Molecules from SMILES-ordered atoms with sinusoidal encodings (left) are coherent and valid, while random ordering (right) produces fragmented, implausible structures, showing SMILES' inductive bias for local structure during early denoising

atoms in a molecule. To test this, we sample from two 15M parameter TABASCO-models, one trained with sinusoid encodings and one without any encodings. We partially noise molecules to different $t \in [\tau, 1)$ and denoise these molecules using the models starting from that point. Figure 4 shows how as $\tau$ increases, the performance difference of the models decreases and switches near the end of denoising. This suggests that the sampling trajectories in the model with positional encodings differ from those the model is trained on (see Appendix B). Furthermore, the higher POSEBUSTERS validity of the positional-encoding-free model towards the end of denoising suggests that in later stages of denoising its sampling trajectory is well aligned with training trajectories. Figure 4 also indicates that the permutationally invariant model is able to create high quality molecules when the final atom positions become apparent from its noisy coordinates. This implies that positional encodings may be especially important in helping distinguish atoms at high noise levels when coordinates contain little information.

Table 4: Ablation study on TABASCO models trained on GEOM-Drugs showing impact of training without postional encodings, batch augmentations or random rotations.

| Configuration | Validity ↑ | Novelty ↑ | Diversity ↑ | PoseBusters ↑ |
|---|---|---|---|---|
| TABASCO-hot (15M) | 0.98 | 0.93 | 0.89 | 0.91 |
| w/o positional encoding | 0.93 | 0.93 | 0.91 | 0.70 |
| w/o batch augmentations | 0.98 | 0.94 | 0.88 | 0.89 |
| w/o random rotations | 0.98 | 0.94 | 0.89 | 0.90 |

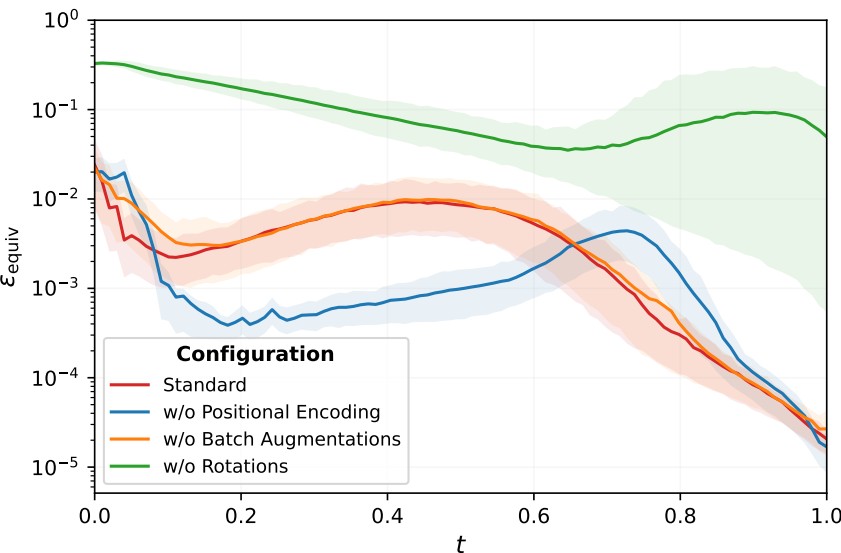

Figure 6: Relative equivariance error across noise levels for TABASCO-hot (15M) when trained without positional encodings, batch augmentations or random rotations.

### 4.6 Analyzing Equivariant Behaviour

**Quantifying the Equivariance Error.** We evaluate the quality of TABASCO's equivariance, as this is not encoded into the architecture. Similarly to previous work, we measure the deviation of the models prediction under random rotations (Karras et al., 2021; Bouchacourt et al., 2021). To control for numerical inaccuracies during sampling, rather than measuring the equivariance of fully denoised molecules, we measure the equivariance of the endpoint predictions of the model by partially noising molecules to different $t \in [0, 1]$. Given noisy coordinates $\mathbf{x}_t$ at time $t$, a random rotation $R$, and a function that at any timestep predicts the endpoint $\hat{\mathbf{x}}_1 = f(\mathbf{x}_t, t)$, we randomly rotate the input and apply the inverse rotation to the output, i.e. $Z(\mathbf{x}_t, t, R) = R^\top f(R\mathbf{x}_t, t)$. We estimate the relative equivariance error with

$$\epsilon_{\text{equiv}} = \text{Var}_R \left[ \frac{Z(\mathbf{x}_t, t, R)}{||Z(\mathbf{x}_t, t, R)||} \right] . \tag{7}$$

We normalize the endpoint prediction per atom within random rotations of the same molecule to account for changes in scale during the sampling process and differing vector magnitudes. In an equivariant model one would have $Z(\mathbf{x}_t, t, R) = f(\mathbf{x}_t, t)$, which would trivially yield $\epsilon_{\text{equiv}} = 0$.

**Ablating the Equivariance Error.** During standard model training we approximately enforce equivariance by randomly rotating the input data and further augmenting the training batch with additional rotations of the same samples. We study the equivariant behaviour of the model by training four 15M TABASCO-hot models with slight modifications: (i) Standard model training, (ii) We train the model without positional encodings, (iii) We apply independent random rotations to all molecules in the batch, but do not augment it with additional random rotations of the same data, (iv) We train the model without any random rotations, i.e. the model always sees the same coordinates for the same molecule. We train the models with the same compute budget as previously allotted: two A100 GPUs over 36 hours.

We compare the observed performance in Table 4 and visualize the equivariance error over time in Figure 6. The results show how in the standard configuration, the model approximates an equivariant function with an error of less than 1% for most $t$, and as denoising progresses the model further reduces the equivariance error of its predictions. The positional encoding-free model approximates equivariance similarly well, however, we observe differences of up to an order of magnitude compared the standard model at certain noise scales.

Simultaneously, randomly rotating data, but omitting intra-batch augmentations with further random rotations, does not worsen the equivariance error, but slightly hurts POSEBUSTERS performance. From this we conclude that intra-batch augmentations do not improve model equivariance, but may improve training, possibly because of higher-quality gradient steps induced by the random rotations.

Finally, Figure 6 and Table 4 show how completely omitting any random rotations during training leads to a model with a high POSEBUSTERS score that has a significantly higher equivariance error than all other models, often larger than 10%. From this we conclude that, while (approximate) equivariance is often a desirable property, our generative model does not require any notion equivariance to generate physically plausible molecules.

## 5 Physically Constrained Last-Mile Pose Guidance

Existing 3D molecule generators yield globally sound conformations but struggle with local stereochemical checks such as POSEBUSTERS. Most violations stem from coordinate drifts near the end of the sampling trajectory $(t \to 1)$. We therefore frame pose refinement as a *last-mile* problem and introduce a lightweight, differentiable guidance step that enforces simple physical distance bounds without force-field evaluation or relaxation.

**Distance–bounds matrix.**   For every atom pair we pre-compute lower and upper bounds $\big[ L_{ij}, U_{ij} \big]$ over 1–5 bond separations, analogously to how POSEBUSTERS computes bounds on valid bond lengths and angles using RDKIT:

- **Lower bound $L_{ij}$:** sum of van-der-Waals radii minus 0.1 Å;

- **Upper bound $U_{ij}$:** cumulative covalent bond lengths along the shortest path in the molecular graph.

These numbers match the Universal Force Field (UFF) limits but are looked up from a static table. Here no UFF energy computation is performed.

**Two-phase sampling with distance-bounds guidance.**

1. **Free denoising.** Run the standard sampler until $t = 0.99$, obtaining noised conformation $(\mathbf{x}_{0.99}, a_{0.99})$.

2. **Guided refinement.** In each remaining denoising step, convert the endpoint predicted coordinates to an RDKIT conformer and look up the physical bounds on atom pair distances $[L_{ij}, U_{ij}]$ for each distance pair $d_{ij} = ||\mathbf{x}_{t,i} - \mathbf{x}_{t,j}||$. The loss on physical constraints is computed with

$$\mathcal{L}_{\mathrm{phys}}(\mathbf{x}_t) = \sum_{i<j} \begin{cases} \big(d_{ij} - U_{ij}\big)^2, & d_{ij} > U_{ij}, \\ \big(d_{ij} - L_{ij}\big)^2, & d_{ij} < L_{ij}, \\ 0, & \text{otherwise}. \end{cases}$$

We back-propagate through the network and apply one gradient step to the inputs:

$$\mathbf{x}_t \ \leftarrow \ \mathbf{x}_t - \alpha_{\mathrm{phys}} \frac{\partial \mathcal{L}_{\mathrm{phys}}}{\partial \mathbf{x}_t}$$

If the molecule decoded at $t = 0.99$ is not RDKIT-valid, no guidance is applied to the sample.

### 5.1 Effect of Physically-Constrained Guidance

In the last row of Table 2 we compare TABASCO with guidance to all evaluated baseline models. In Table 5 we compare physically-constrained guidance to UFF relaxation of unguided molecules: In all experiments

Table 5: Effect of distance-bounds guidance on PoseBusters validity and runtime (single A100) on TABASCO-hot (15M) for 1000 molecules. RMSD is evaluated with respect to the unguided baseline. $^{\gamma}$UFF calculations were performed on an M3 MacBook Pro.

| Method | PoseBusters↑ | Strain Energy ↓ | Diversity ↑ | RMSD | Runtime ↓ |
|---|---|---|---|---|---|
| Baseline | 0.91 | 14.16 | 0.88 | - | 10.67 |
| w/ UFF$^{\gamma}$ | 0.94 | 4.74 | 0.88 | 0.226 | 14.21 |
| w/ Constr. UFF$^{\gamma}$ | 0.93 | 11.15 | 0.88 | 0.084 | 23.42 |
| w/ guidance | 0.94 | 19.23 | 0.89 | 0.132 | 75.65 |

the same molecules are denoised identically up to $t = 0.99$. In one experiment we allow for unconstrained relaxation and in another introduce a movement constraint of 0.1Å on each atoms original location. We choose $\alpha_{phys} = 0.01$ in all experiments. Table 5 shows how distance-bounds guidance improves PoseBusters validity, while preserving diversity and slightly increasing strain energy. Although distance-bounds guidance increases sampling time due to sequential bound computation and backpropagation, overall sampling remains faster than in prior approaches. The method also largely preserves atom positions compared to unguided baselines and preserves diversity, since the molecular hypothesis is mostly fixed by $t = 0.99$ and the model is only guided to create a more physically plausible conformer of the same molecule. We highlight that in practice, our last-mile guidance and UFF target different objectives – physical plausibility vs. energy minimization – and that guidance is superior in that it yields better PoseBusters validity while better retaining the model's predicted conformation. In Appendix E we provide a more extensive comparison of model performance across parameter scales with and without guidance. The approach is model-agnostic and applies to any diffusion- or flow-based 3D generator that exposes gradients with respect to atom coordinates. Unlike force-conditioned samplers such as DiffForce (Kulytė et al., 2024), which require full molecular-mechanics gradients at every reverse step, our method uses pre-tabulated distance bounds and needs no energy evaluation or additional learnable parameters.

## 6 Discussion

Our findings align with recent trends toward simpler architectures: Although SE(3) equivariance is often considered essential, our non-equivariant model learns equivariant representations up to small errors and achieves state-of-the-art performance on physical plausibility benchmarks, suggesting that enforced symmetries may be restrictive for some generation tasks. Our theoretical understanding of the approximate equivariance remains limited, but we are able to verify that empirically the model exhibits very low equivariance error. Our stripped-down architecture is easily extensible and only models coordinates and atom types explicitly and encodes atoms with sequential positional encodings. Conversely, omitting explicitly modelled bonds can limit conditioning when aiming to enforce valences or bond types (Peng et al., 2024), and encoding molecules with SMILES-based positional encodings may introduce systemic biases. Simultaneously, our model yields a ten-fold speed improvement compared to previous methods, potentially making practical applications like large-scale virtual screening more feasible in the future. We also observe that only minor performance improvements emerge at scale unlike for previous work on other molecular modalities (Abramson et al., 2024; Geffner et al., 2025), which may point to the necessity of larger training budgets or may indicate the model is close to saturating the training dataset PoseBusters-validity of 0.94. Physically-constrained guidance is shown to be effective for improving physical plausibility with minimal modifications, however compared to traditional methods like UFF Relaxation, it is based on chemoinformatics heuristics, remains more expensive and converges to higher strain energies.

We provide a more detailed analysis of the models limitations in Appendix G.

## 7   Conclusion

In this work we present TABASCO, a non-equivariant generative model for 3D small molecule design that exhibits enhanced scalability and performance on physical plausibility compared to baselines. We study the importance of positional encodings for small molecules, investigate the emergent equivariant properties of our model and the effects of scaling the model to large sizes. We hope that our model serves as a compelling example of how minimalist architectures can be effectively applied to molecular design and that our code base acts as an extensible tool for integration in drug-discovery workflows, for example through conditioned generation on relevant modalities or RL-based property optimization.

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

## A  Comparison to Previous Work

For all compared baselines we sample 1000 molecules with three random seeds on an A100 GPU. We report averages over the three runs.

**EQGAT-diff**  We evaluated EQGAT-diff using the official codebase on GitHub[1] and the checkpoints linked there. We used the example evaluation script, which we edited to save molecules as outputted from reverse sampling, without any post-processing.

**FlowMol**  We used the official implementation for FlowMol and the linked checkpoints on GitHub[2] and sampled molecules using the default script. For both GEOM[3] and QM9 (Ramakrishnan et al., 2014) we benchmarked againts the CTMC-based models.

**Semla Flow**  We evaluated SemlaFlow using the sampling script and model checkpoints from GitHub[4]. We modified the sampling script to save all outputs from the model, as opposed to only valid molecules.

**ADiT**  We benchmark ADiT by evaluating the molecules provided in the paper's GitHub repository, accessed on June 28th 2025[5]. When directly generating molecules from the ADiT model checkpoint[6] we observed worse values on all metrics. A likely explanation for this is taht the publicly available checkpoints are from an independent reproduction and do not correspond to the reported values in the paper, as stated in the public repository. As a result, we opt for the metrics computed from the publicly available molecules.

## B  Ablations

**Time Distribution During Training.**  Based on success in previous works, we choose the Beta-distribution for sampling the time $t$ during training (Irwin et al., 2025; Geffner et al., 2025). We investigate the effect of different $\alpha$ values for the training time distribution $\text{Beta}(\alpha, 1)$ and ablate three values in Table 6. We observe significant changes in performance at sampling time when shifting the probability weight assigned to different times $t \in [0, 1]$ during training, and empirically find that $\text{Beta}(1.8, 1)$ yields the best results for our purpose.

**Stochasticity Ablation.**  We investigate several choices for $g(t)$ in Eq. 6 based on the approach in Geffner et al. (2025). Throughout this work, except when explicitly stated otherwise, we set $g(t)$ to zero for $t > 0.9$ to allow for precise placement of atoms towards the end of sampling. Table 7 compares the effect of four possible stochasticity functions. We observe that except for $g(t) = 0$ performance is very similar across all metrics. We trace the contrast between $g(t) = 0$ and the other parameterizations to a difference in sampling trajectories and show a comparison in Figure 7. In contrast to the trajectory of $g(t) = 0$ which is consistent

---

[1] `https://github.com/jule-c/eqgat_diff/`, MIT License
[2] `https://github.com/Dunni3/FlowMol/tree/8f777fd57c9e1decc4f9ef6a76b366dac874c838`, MIT License
[3] Axelrod, Simon, et al. "GEOM, energy-annotated molecular conformations for property prediction and molecular generation." *Sci Data* 9, 185 (2022). Available under CC0 1.0.
[4] `https://github.com/rssrwn/semla-flow/`, MIT License
[5] https://github.com/facebookresearch/all-atom-diffusion-transformer
[6] https://huggingface.co/chaitjo/all-atom-diffusion-transformer

Table 6: Ablation of $\alpha$ in the training time $t$ distribution $\text{Beta}(\alpha, 1)$ on TABASCO-hot (15M) trained on GEOM-Drugs.

| $\alpha$ | Validity ↑ | Novelty ↑ | Diversity ↑ | PoseBusters ↑ |
|---|---|---|---|---|
| 1.5 | 0.96 | 0.92 | 0.89 | 0.84 |
| 1.8 | 0.98 | 0.93 | 0.88 | 0.91 |
| 2.0 | 0.97 | 0.93 | 0.88 | 0.89 |

Table 7: Effect of four possible $g(t)$ parameterizations on TABASCO trained on GEOM-Drugs. For all $t > 0.9$ we set $g(t) = 0$ and use $\epsilon = 0.01$.

| $g(t)$ | Validity ↑ | Novelty ↑ | Diversity ↑ | PoseBusters ↑ |
|---|---|---|---|---|
| $0$ | 0.96 | **0.95** | 0.90 | 0.83 |
| $\frac{1}{t+\epsilon}$ | **0.98** | 0.93 | 0.88 | **0.91** |
| $\frac{1}{t^2+\epsilon}$ | 0.97 | 0.93 | 0.89 | **0.91** |
| $\frac{1-t}{t+\epsilon}$ | **0.98** | 0.94 | 0.89 | 0.89 |
| *w/o positional encodings* | | | | |
| $0$ | 0.94 | 0.93 | **0.91** | 0.69 |
| $\frac{1}{t+\epsilon}$ | 0.89 | 0.87 | **0.91** | 0.26 |

with the training trajectory, the rest of the $g(t)$ functions have very large magnitudes close to $t = 0$, which empirically leads first to an explosion and then to a collapse of the atom vector magnitudes. In the collapsed state atoms are roughly arranged in a sequence and slowly grow into the finished molecule as $t \to 1$ (see Figure 7). This sudden rearrangement and growing into the finished molecule appears to yield better final molecules compared to when following the training trajectories more closely. This may also help explain the dip in POSEBUSTERS validity during partial molecule noising of TABASCO with positional encodings in Figure 4. In contrast to this, we observe in Table 7 that this explosion and collapse behaviour leads to much worse molecules when positional encodings are not added to the model, possibly because in the collapsed state atom coordinates are almost identical and become very hard to distinguish without positional encodings.

**Number of Steps at Sampling.**  We investigate the effect of reducing the number of sampling steps on molecule quality and ablate over several choices in Table 8. We observe that as little as 40 steps are necessary for TABASCO-hot to outperform previous methods on POSEBUSTERS. We further observe that additional steps have no effect on molecular quality.

Table 8: Number of steps at sampling for TABASCO-hot (15M) trained on GEOM-Drugs. We additionally evaluate connectivity, which denotes the fraction of fully connected molecules.

| # Steps | Validity ↑ | Novelty ↑ | Connectivity ↑ | PoseBusters ↑ |
|---|---|---|---|---|
| 10 | 0.99 | 0.98 | 0.00 | 0.00 |
| 20 | 1.00 | 0.99 | 0.00 | 0.00 |
| 30 | 0.98 | 0.97 | 0.99 | 0.81 |
| 40 | 0.98 | 0.94 | 0.99 | 0.91 |
| 50 | 0.99 | 0.95 | 1.00 | 0.91 |
| 100 | 0.98 | 0.94 | 1.00 | 0.91 |
| 200 | 0.98 | 0.93 | 1.00 | 0.89 |
| 500 | 0.98 | 0.94 | 1.00 | 0.91 |

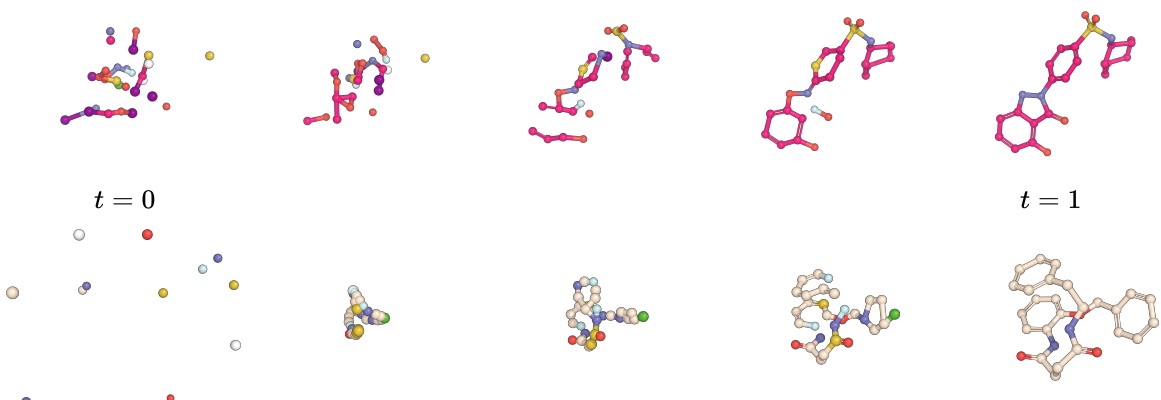

Figure 7: Snapshots of the sampling trajectories for two different molecules sampled from TABASCO-hot (15M) trained on GEOM-Drugs. The upper trajectory is sampled with $g(t) = 0$ and the lower one with $g(t) = \frac{1}{t+0.01}$

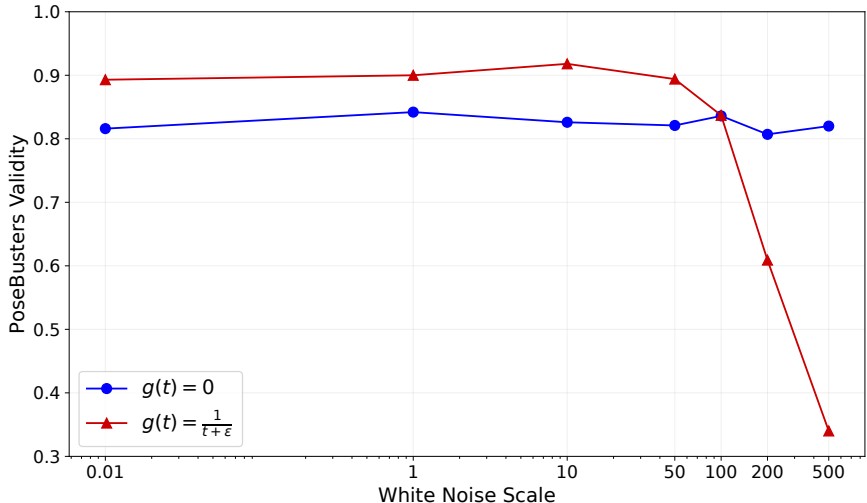

Figure 8: Comparison of POSEBUSTERS-validity across noise scales $\gamma$ with different $g(t)$. In contrast to all other comparisons we set $g(t) = 0$ only beyond $t > 0.95$ to further augment the effect of adding noise.

**Noise Scaling.** We investigate several values for $\gamma$ to ablate the effect of noise scaling. In Figure 8 we compare POSEBUSTERS-validity for different noise scales and different $g(t)$ parameterizations. We observe that molecular quality remains high over several noise scales, and then collapses for $g(t) = \frac{1}{t+\epsilon}$.

**Training Components.** We ablate three training components in Table 9 for the 15M-parameter TABASCO-hot model on GEOM-Drugs. Removing weight-EMA has almost no effect: all headline metrics change by $\leq 0.01$ and diversity rises slightly. This shows that the model does not rely on EMA for chemical or geometric correctness. Performance is more sensitive to the coordination between input and output coordinate features. Eliminating the single cross-attention block lowers validity and novelty by $\sim 0.04$ and, critically, drops POSEBUSTERS validity to 0.80. This indicates that coupling atom-type and coordinate information is necessary to resolve physical validity constraints such as steric clashes and strain at this parameter scale. Positional encodings also prove critical, as investigated in Section 4.5. Without them, raw validity remains high (0.93) but POSEBUSTERS validity collapses to 0.70, revealing widespread geometric artifacts. The model

can still generate chemically plausible graphs, yet struggles to arrange them in physically realistic 3D space (see Figure 5).

Table 9: Ablation study of model performance when removing components. Layer counts are adjusted to match model size where needed. Higher values are better.

| Method | Validity | Novelty | Diversity | PoseBusters |
|---|---|---|---|---|
| TABASCO-hot | 0.98 | 0.93 | 0.88 | 0.91 |
| w/o EMA | 0.98 | 0.93 | 0.89 | 0.91 |
| w/o cross-attention | 0.94 | 0.89 | 0.89 | 0.80 |
| w/o positional encoding | 0.93 | 0.93 | 0.91 | 0.70 |

## C  Jensen-Shannon Divergence Comparison

Table 10: Jensen-Shannon Divergence on several extracted molecule features. For each method 3,000 molecules were tested on GEOM-Drugs. Lower is better on all metrics.

| Model | Bond Length | Bond Angle | Dihedral | Freq. Bond Pair | Freq. Bond Triplet | Bonds per Atom | Num. Rings | Bond Type |
|---|---|---|---|---|---|---|---|---|
| SemlaFlow | 0.4085 | 0.1502 | **0.0673** | 0.0677 | 0.0664 | 0.1044 | 0.1135 | 0.0384 |
| EQGAT-diff | **0.2334** | **0.1178** | 0.0924 | **0.0440** | **0.0507** | 0.0728 | **0.0677** | **0.0124** |
| FlowMol | 0.3060 | 0.1697 | 0.1399 | 0.1233 | 0.1202 | 0.1709 | 0.1350 | 0.0889 |
| TABASCO-hot | 0.3589 | 0.1538 | 0.0892 | 0.0614 | 0.0737 | **0.0713** | 0.0803 | 0.0419 |

In Table 10 we evaluate the JSD of several molecule features between the generated and GEOM-Drugs molecules for TABASCO-hot (15M) and three baselines. We observe only minor differences in the JSD of features between all tested methods. EQGAT-diff most closely matches the molecule distribution of GEOM-Drugs on the evaluated features, however, improvements in JSD do not appear clearly to correlate with performance improvements in physical validity (Table 1) or other metrics (Table 2). This may indicate that the JSD estimate is too noisy or the distance to the GEOM-Drugs distribution is too large across all features for improvements to give meaningful insights on the quality of generated molecules.

## D  Extended Details on Models

In this section we further elaborate on model architecture (Figure 2), give concrete values for relevant hyperparameters in Table 11, and describe the unconditional sampling algorithm in detail in Algorithm 1. Atom coordinates are encoded with a bias-free linear layer that scales to the model's hidden size. Discrete atom types are encoded through an embedding layer, where we model Carbon, Nitrogen, Oxygen, Fluorine, Sulfur, Chlorine, Bromine, Iodine and a miscellaneous "$*$" atom, for all elements in the training set not contained within the previous list. We encode the time $t \in [0, 1]$ through a Fourier encoding, and each atoms location within the molecules's SMILES sequence through a standard sinusoid encoding. We tried concatenating these four vectors and creating a combined hidden representation with an MLP mapping from $\mathbb{R}^{4 \times \text{hidden dim}} \rightarrow \mathbb{R}^{\text{hidden dim}}$ but observed no difference in practice to simply adding the vector representations, and thus opted for this approach. Each transformer block applies layer-norm to the activations, then PyTorch multi-head attention, another layer-norm and a transition layer, where we include residual connections between the first two and second two components. This output is processed by two parallel PyTorch cross-attention layers one for atom types and for coordinates. Each consists of a self-attention block, a multi-head attention block, where the original combined hidden representation is used as key and value, and a feed-forward block. Both outputs are subsequently processed through domain-specific MLPs where the output atom coordinate MLP is also bias-free.

Table 11: Model hyperparameters across different model sizes

| Hyperparam. | TABASCO-mild | TABASCO-hot | TABASCO-spicy |
|---|---|---|---|
| # Params. | 3.711.369 | 14.795.529 | 59.082.249 |
| Hidden size | 128 | 256 | 512 |
| # Transformer blocks | 16 | 16 | 16 |
| # Attn. heads | 8 | 8 | 8 |
| Train $t$ distribution | Beta(1.8,1) | Beta(1.8,1) | Beta(1.8,1) |
| $\lambda_{\text{discrete}}$ | 0.1 | 0.1 | 0.1 |
| Learning rate | 0.001 | 0.001 | 0.0005 |
| Optimizer | Adam | Adam | Adam |
| EMA-weight | 0.999 | 0.999 | 0.999 |
| Batch size | 256 | 256 | 128 |
| # Rotation Augs. | 8 | 8 | 8 |
| Effective batch size | 2048 | 2048 | 1024 |
| # GPUs | 2 | 2 | 2 |
| Training Duration | 36h | 36h | 72h |
| # Sampling Steps | 100 | 100 | 100 |
| $g(t)$ | $\frac{1}{t+0.01}$ | $\frac{1}{t+0.01}$ | $\frac{1}{t+0.01}$ |
| $\gamma$ | 0.01 | 0.01 | 0.01 |

Table 12: Comparison POSEBUSTERS validity when adding physically-constrained guidance. Evaluated on 1000 molecules on a single A100 GPU.

| Method | # Params. | Validity ↑ | Novelty ↑ | Diversity ↑ | PoseBusters ↑ | Strain Energy ↓ | Time ↓ (s) |
|---|---|---|---|---|---|---|---|
| TABASCO-mild | 3.7M | 0.95 | 0.93 | 0.89 | 0.85 | 21.32 | 5.9 |
| *w/ guidance* | | 0.96 | 0.95 | 0.89 | 0.91 | 26.53 | 60.86 |
| TABASCO-hot | 15M | 0.98 | 0.93 | 0.88 | 0.91 | 14.16 | 10.67 |
| *w/ guidance* | | 0.97 | 0.94 | 0.89 | 0.94 | 19.23 | 75.66 |
| TABASCO-spicy | 59M | 0.97 | 0.90 | 0.89 | 0.92 | 15.07 | 19.77 |
| *w/ guidance* | | 0.97 | 0.93 | 0.89 | 0.94 | 17.01 | 131.80 |

# E   Further Analysis on Physical Guidance

We provide a detailed description of our physically constrained guidance procedure in Algorithm 2 and provide full results on all model sizes in Table 12. We observe the largest improvement in POSEBUSTERS for TABASCO-small, and minor improvements in novelty for all model sizes. Simultaneously we consistently observe an increase in strain energy when applying physical guidance.

# F   Failure Case Analysis

We study common physical validity failure modes of generated molecules using TABASCO-hot (15M). Figure 10 shows the phi coefficient matrix, quantifying pairwise correlation between subtest failures. Table 1 shows the absolute failure rate of each subtest. Unsurprisingly, bond length and bond angle violations are strongly correlated, since when local geometry is implausible, both properties tend to be affected. These geometric failures also correlate with internal energy violations, likely because distorted bonds and angles lead to higher strain. The most frequent failure mode is steric clashes (Table 1), however, this does not correlate strongly with other tests. We hypothesize this is because clashing atom pairs are not bonded, so for example bond-specific tests do not apply to them.

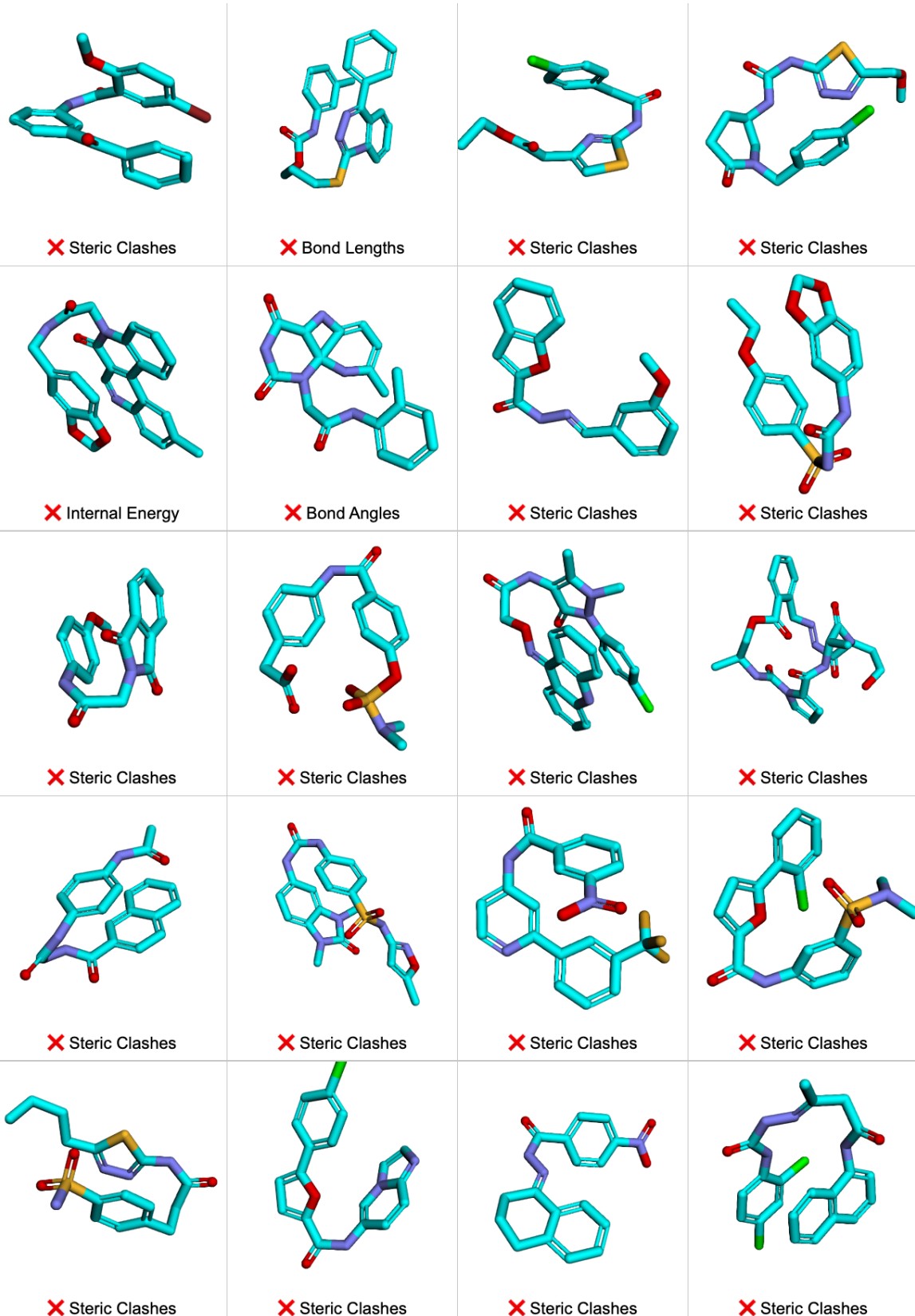

Figure 9: Twenty generated molecules chosen at random that do not satisfy physical plausibility, each labeled with the failed subtest.

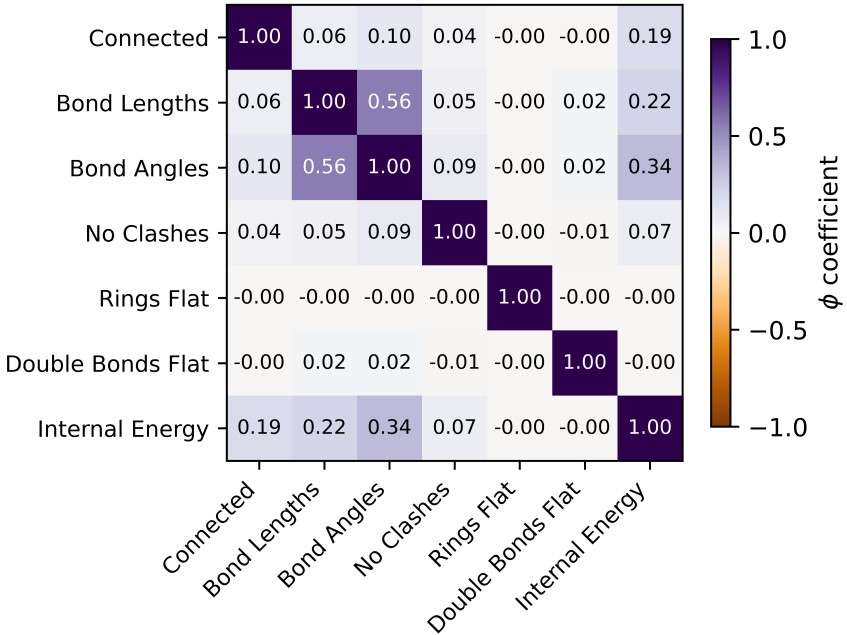

Figure 10: Phi coefficient matrix of physical validity subtests.

In Figure 9 we show a random subset of failing molecules along with the test that failed. We observe that as shown in Table 1 steric clashes are by far the most common failure mode. We also note that while several molecules are visibly flawed, in some cases the violation of physical validity is not immediately apparent. This further highlights the need for rigorous physical plausibility tests such as POSEBUSTERS.

## G Limitations

The approach described in this work retains several limitations. SMILES-derived positional encodings improve performance but can theoretically introduce systemic biases, that may limit the model when faced with unusual bond patterns or non-standard chemical structures. Furthermore, omitting explicit bond modeling creates a leaner model and simpler training objective, but limits control over valences and bond orders when sampling the model. While TABASCO exhibits emergent equivariance, in some areas such as molecular dynamics, where even small equivariance errors can prove problematic, this approximate equivariance may still be insufficient.

We also observe a scaling plateau: performance improvements diminish significantly beyond the 15M parameter scale, with only modest gains observed when scaling to 59M parameters. This suggests that further scaling benefits may require increasingly larger training compute budgets. Alternatively, this plateau may be partially by the training dataset imposing a glass ceiling on physical validity which is almost saturated by the 15M parameter model (see Table 1). Understanding the precise factors underlying this plateau remains an open question.

While we demonstrate that TABASCO learns approximately equivariant representations through data augmentation alone, our mechanistic understanding of this emergent equivariance remains limited. We quantify the equivariance error empirically (Section 4.6), but do not provide a theoretical characterization of how the model internally represents rotational symmetry or why certain training configurations yield lower equivariance error than others. Further investigation into the learned representations could yield insights for

designing more principled non-equivariant architectures.

The validation presented in this work focuses exclusively on unconditional molecular generation. While the architecture is designed to be readily extensible to conditional tasks such as property-guided generation or structure-based drug design, we have not empirically validated performance in these settings. The benefits observed for unconditional generation may not directly transfer to conditional scenarios, which often require different architectural considerations and evaluation protocols.

The physically-constrained guidance algorithm serves as a proof-of-concept for boosting the physical plausibility of molecules during sampling without requiring any modifications to training data, model architecture or parameter scale. Still, this approach is based on optimizing chemoinformatics heuristics for high-quality molecules and it dramatically increases sampling times.

Furthermore, while useful to quantify physical plausibility of 3D molecules, POSEBUSTERS cannot capture all aspects of molecular quality, and does not quantify additional very relevant metrics of interest: TABASCO does not address improvements in drug-likeness of molecules or synthetic accessibility.

---

**Algorithm 1** Unconditional Sampling Algorithm

---

**procedure** EUCLIDEANSTEP$(\mathbf{x}_t, \hat{\mathbf{x}}_1, t, \Delta t, g(\cdot), \gamma)$
    $\mathbf{v}_t \leftarrow \frac{1}{1-t}(\hat{\mathbf{x}}_1 - \mathbf{x}_t)$
    $\mathbf{s}_t \leftarrow g(t)\frac{t\mathbf{v}_t - \mathbf{x}_t}{1-t}$
    $\mathrm{d}W_t \leftarrow \sqrt{2\,\gamma\,g(t)}\,\mathcal{N}(0, I)$
    $\mathbf{x}_t \leftarrow (\mathbf{v}_t + \mathbf{s}_t + \mathrm{d}W_t)\Delta t$
    **return** $\mathbf{x}_t$
**end procedure**

**procedure** DISCRETEFLOWSTEP$(a_t, \hat{p}_1, t, \Delta t)$         ▷ All indexed ops without loops are vectorized
    $\mathbf{r}_t(i, \cdot) = \frac{\Delta t}{1-t}\hat{p}_1(i)$         ▷ $\hat{p}_1$ consists of softmax-normalized model logits
    $\mathbf{r}_t(i, a_t(i)) \leftarrow -\sum_{j \neq a_t(i)} \mathbf{r}_t(i, j)$         ▷ Make $\mathbf{r}_t$ zero mean
    $\mathbf{p}_{t+\Delta t}(i, j) \leftarrow \mathbf{1}_{a_t(i)=j} + \mathbf{r}_t(i, j)$
    $a_t(i) \leftarrow \mathrm{Categorical}(\mathbf{p}_{t+\Delta t}(i, \cdot))$
    **return** $a_t$
**end procedure**

**procedure** SAMPLEMOLECULE$(f, \{t_i\}_{i=0}^N)$
    $\mathbf{x} \leftarrow \mathcal{N}(0, I)$
    $a \leftarrow \mathrm{Categorical}\left(\delta(\frac{1}{\#\text{ atom types}})\right)$
    **for** $i = 1$ **to** $N$ **do**
        $\Delta t \leftarrow t_i - t_{i-1}$
        $(\hat{\mathbf{x}}_1, \hat{p}_1) \leftarrow \mathrm{EndpointPrediction}(f, (\mathbf{x}, a), t_i)$    ⎫
        $\mathbf{x} \leftarrow \mathrm{EUCLIDEANSTEP}(\mathbf{x}_t, \hat{\mathbf{x}}_1, t_i, \Delta t)$     ⎬ SAMPLINGSTEP
        $a \leftarrow \mathrm{DISCRETEFLOWSTEP}(a_t, \hat{p}_1, t_i, \Delta t)$    ⎭
    **end for**
    **return** $(\mathbf{x}, a)$
**end procedure**

---

---

**Algorithm 2** Flow Matching with Physical Guidance

---

1: **procedure** PHYSICALGUIDANCE($f, (\mathbf{x}_t, a_t), t, \alpha$)
2:     $\hat{\mathbf{x}}_1, \hat{p}_1 \leftarrow \text{EndpointPrediction}(f, (\mathbf{x}_t, a_t), t)$
3:     $\hat{a}_1(i) = \text{argmax}_j \ \hat{p}_1(i, j)$
4:     bounds $\leftarrow \text{GetPhysicalConstraints}(\hat{\mathbf{x}}_1, \hat{a}_1)$                    ▷ Calls RDKIT `GetBoundsMatrix()`
5:     **for** each atom pair $(\mathbf{x}_{t,i}, \mathbf{x}_{t,j})$ in $\mathbf{x}_t$ **do**                    ▷ This nested loop is vectorized in practice
6:         $d_{ij} \leftarrow ||\mathbf{x}_{t,i} - \mathbf{x}_{t,j}||_2^2$
7:         **if** $d_{ij} < \text{bounds}_{ij}^{\min}$ **then**                    ▷ Can also regress towards the interval centre
8:             $\mathcal{L} \leftarrow \mathcal{L} + (d_{ij} - \text{bounds}_{ij}^{\min})^2$
9:         **else if** $d_{ij} > \text{bounds}_{ij}^{\max}$ **then**
10:            $\mathcal{L} \leftarrow \mathcal{L} + (d_{ij} - \text{bounds}_{ij}^{\max})^2$
11:        **end if**
12:    **end for**
13:    $\mathbf{x}_t \leftarrow \mathbf{x}_t - \alpha \cdot \text{sign}(\nabla_{\mathbf{x}_t} \mathcal{L})$                    ▷ The sign-op slightly stabilizes updates in practice
14:    **return** $\mathbf{x}_t$
15: **end procedure**

16: **procedure** GUIDEDSAMPLING($f, (\mathbf{x}_0, a_0), \{t_i\}_{i=0}^{N}, t_{\text{guidance}}$)
17:    $(\mathbf{x}, a) \leftarrow (\mathbf{x}_0, a_0)$
18:    **for** $i = 1$ **to** $N$ **do**
19:        $\Delta t \leftarrow t_i - t_{i-1}$
20:        **if** $t_i \geq t_{\text{guidance}}$ **then**
21:            $\mathbf{x} \leftarrow \text{PHYSICALGUIDANCE}(f, (\mathbf{x}, a), t_i, \alpha)$
22:        **end if**
23:        $(\mathbf{x}, a) \leftarrow \text{SAMPLINGSTEP}(f, (\mathbf{x}, a), t_i, \Delta t)$
24:    **end for**
25:    **return** $(\mathbf{x}, a)$
26: **end procedure**

---

