# OpenReview forum: "TABASCO: A Fast, Simplified Model for Molecular Generation with Improved Physical Quality"
_TMLR — Accepted by TMLR_

### Review · Reviewer_6mdq · 2025-11-24

**Summary Of Contributions:**

This approach relaxes the widely used special assumptions or designs, equivariance, permutation invariance, and graph structure, in molecule encoding models, resulting a much more simplified architecture while achieving some better properties.

The model is demonstrated to generate molecules with higher physical quality and faster in inference.

The finding of study that even random rotation is not necessary is also interesting.

**Audience:**

Yes

**Audience Explanation:**

Simpler architectures are very important for bio deep learning models to be democratized and are a recent trend in many different sub-fields in molecule-related tasks.

The successful relaxation of the assumptions has a potentially high impact.

**Claims And Evidence:**

Yes

**Claims Explanation:**

The claim of simple architecture is supported by the model design.

The claimed fast and high quality properties are also evident in the experiment results.

**Requested Changes:**

- Critical to securing my recommendation for acceptance:

The motivation to discard molecules with more than 72 heavy atoms from the training dataset is unclear, as transformer is absolutely capable of handling at least hundreds of tokens.

Other widely adopted metrics like RMSD for unconditional small molecule generation should be reported on the proposed model and compared with baselines. RMSD is reported in the current manuscript, but not compared with baselines. I understand it might not be sota in all senses, which is fine, but reporting them would be beneficial for a comprehensive understanding of the advantages and potential disadvantages of the simplication of assumpations.

- Simply strengthen the work in my view:

the letters x and a in fig. 2 should be explained in the figure itself or in the caption.

The intuition toweigh the loss could be discussed.

In fig. 3, the metrics of showcased molecules could be shown to have a better understanding for readers.

The definition of Z(xt, t, R) is missing.

---

> ### Author Response · Authors · 2025-12-13
> **Response 1**
>
> We thank the reviewer for their feedback and suggestions. Below we address each of the concerns:
>
> > motivation to discard molecules with more than 72 heavy atoms
>
> Indeed the transformer architecture is not limiting in terms of the input’s token length. We choose to discard molecules with more than 72 heavy atoms in line with the pre-processing done in previous work [1]. Limiting the number of atoms allows for improved training times by maintaining a significantly larger batch size, and avoiding the memory bottlenecks inherent to calculating dense attention matrices for larger molecules. These represent, however, just 1% of the training data so almost all the diversity is preserved.
>
> > On reporting RMSD
>
> We report RMSD in the physically constrained guidance experiment (Section 5.1), where it measures how much the guided samples deviate from the baseline conformations. This metric is appropriate in that setting because a reference structure is available. However, RMSD is not meaningful for unconditional generation, where no reference molecule exists. We agree that mentioning RMSD in the “Evaluation metrics” section (Section 4.2) is indeed misleading, and we have revised the manuscript to clarify that RMSD is used solely for the guidance experiment and not for our main unconditional generation results.
>
> > the letters x and a in fig. 2 should be explained in the figure itself or in the caption
>
> We have revised the manuscript to clarify these notations.
>
> > The intuition to weigh the loss could be discussed
>
> Our training objective combines two forms of loss weighting: a modality-specific coefficient, $\lambda_{\text{discrete}}$, and a time-dependent weighting schedule. The former is treated as a tunable hyperparameter, while the latter was introduced by Salimans et al. [2] and later applied to molecular design by Le et al. [3]. We emphasize that increasing the loss weight as $t \to 1$ encourages the model to place atomic coordinates with high precision, since the backbone model approaches the identity mapping near the endpoint. We have added the appropriate citations in the revised manuscript.
>
> We are grateful for the reviewer’s engaged feedback and are glad to elaborate further or make any necessary corrections.
>
>
> [1] Ross Irwin et al., SemlaFlow -- Efficient 3D Molecular Generation with Latent Attention and Equivariant Flow Matching, ICLR 2025
> [2] Tim Salimans et al., Progressive distillation for fast sampling of diffusion models, ICLR 2022
> [3] Tuan Le et al., Navigating the design space of equivariant diffusion-based generative models for de novo 3d molecule generation, ICLR 2024

---

### Review · Reviewer_Hc1X · 2025-12-03

**Summary Of Contributions:**

The authors present TABASCO, a molecule generative model based on non-equivariant Transformer architecture. TABASCO co-generates atomic coordinates as well as atomic types through continuous and discrete flow-matching, respectively. The authors also propose physical constrained for last-mile pose guidance to improve physical validity in generated molecules. Finally, the physically-constrained refinement step is introduced to boost the physical realism of the generation. Experimental results on GEOM-DRUGS demonstrated competitive performance in molecular generation.

I find this work being a strong validation of non-equivariant architectures in 3D molecular generation. It is nice to see a standard Transformer, trained via flow-matching, achieve competitive performance with minimal inductive biases. This simpler approach surpasses curated equivariant methods and offers better inference speed.

**Audience:**

Yes

**Audience Explanation:**

The work is closely related to AI4Science community which builds a generative model for molecular generation. It also follows the recent trend of building architecture without strong inductive bias which is mostly supported by the experimental results. I believe the findings shown in this work will benefit the community in building more powerful generative models to solve scientific problems.

**Claims And Evidence:**

Yes

**Claims Explanation:**

The experimental results support the paper's argument that a simple and non-equivariant architectures can be effective for 3D molecular generation while being efficient. TABASCO utilizes a standard Transformer trained via flow-matching without graph-based message-passing or equivariant layers. Despite its relative simplicity, the model surpasses more complex / equivariant baselines on the GEOM-Drugs benchmark, and show efficiency in both sampling speed. Furthermore, the paper also includes ablation studies regarding positional encoding and cross-attention which validates some of the core design choices.

**Requested Changes:**

1. I would like to see how the proposed method works on conditional generation, e.g., generating molecules conditioned on certain properties / pockets. Conditional generation is closely related to the practical applications of molecular generation which would further showcases the effectiveness of TABASCO.
2. The model relies on positional encoding of SMILES, but will it introduce ordering bias that misalign with generated 3D geometries? Will it help if other positional encodings like pairwise distances between atoms are applied?
3. Given the model doesn't force physical constraints, are there any failure cases spotted in inference? And are there certain modes in the failure cases?
4. In Table 5, it is shown that UFF outperforms distance-bounds guidance on speed and strain energy. What are the advantages of using the distance-bounds guidance instead of UFF?
5. How is the proposed method extendable to larger scale like biomolecules? Do the authors think there're fundamental obstacles for the proposed method to work on larger molecular systems?
6. It would also help better compare the performance if standard deviations of main metrics are reported.

---

> ### Author Response · Authors · 2025-12-13
> **Response 1**
>
> We thank the reviewer for the positive and thoughtful assessment of our work. We appreciate the recognition of TABASCO’s contributions—demonstrating that a non-equivariant Transformer trained with continuous and discrete flow matching can achieve competitive 3D molecular generation, benefit from physically constrained guidance, and outperform more complex equivariant models in both quality and efficiency. Below, we address the remaining points in detail.
>
> > 1. Conditional generation
>
> Thank you for the suggestion. Extending TABASCO to conditional generation is a natural next step, and the architecture is designed to support it directly. TABASCO’s modules can accept additional conditioning tokens, such as property embeddings, pocket descriptors, or pharmacophore features, without architectural changes.
>
> We chose, however, to focus our current work on unconditional generation to isolate the behaviour of a minimalist architecture and do a rigorous assessment compared to unconditional baselines. We intend for this version of the TABASCO manuscript to sit alongside and be directly comparable to prior work that focuses exclusively on unconditional generation, ensuring clarity and consistency in how our results are presented [1-4]. We clarify our choices of experiments and the extensibility of the framework in the revised manuscript.
>
> > 2. SMILES-based positional encodings
>
> Our experiments show that although SMILES-based positional encodings introduce a deterministic ordering, they do not harm geometric fidelity; in fact, they significantly improve physical plausibility.
>
> Sections 4.5 and Fig. 4 show that removing positional encodings causes PoseBusters validity to drop from 0.91 to 0.70 (Tab. 4). As discussed there, positional encodings provide useful disambiguation in early high-noise denoising steps where coordinates contain little information. Once coordinates become informative, the model no longer relies on this ordering signal. Moreover, in Fig. 6 we observe that approximate SE(3) equivariance is preserved even with positional encodings. We agree that pairwise-distance or relative-position encodings could be explored. However, they reintroduce geometric features that we intentionally removed to test the limits of minimal inductive bias.
>
> > 3. Physical quality failure cases
>
> Yes—although TABASCO achieves state-of-the-art physical plausibility, we do observe systematic failure modes. We observe local steric issues (clashes, length/angle drift) near t=1, which motivated our last-mile guidance experiments (Section 5). These modes are substantially mitigated by guidance. We added examples and correlation statistics in the appendix in the revised manuscript.
>
> > 4. On distance-bounds guidance vs. UFF
>
> We motivate the choice of distance-bounds guidance over UFF through the higher structural fidelity to the generated sample: the RMSD relative to the unguided sample is lower for guidance (0.132 A) than for unconstrained UFF (0.226 A). That is, UFF more aggressively alters conformations unless it is constrained to retain the original geometry, at which point it suffers a loss in PoseBusters validity. Unlike UFF, our method backpropagates through the generator and aligns the refinement with the learned endpoint prediction, preserving the model’s molecular hypothesis. We clarify these points in the manuscript.
>
>
> > 5. Extensibility to larger biomolecules
>
> Our architecture is, in principle, well suited to scaling toward larger systems. Because TABASCO is based on a standard Transformer operating on sequences of atom-wise embeddings, it can accommodate substantially longer inputs without architectural modification, and the flow-matching objective remains valid regardless of system size. That said, larger biomolecules introduce structural complexities, such as long-range interactions, hierarchical organization etc., and architectural inductive biases of the kind used in AlphaFold3 (e.g., pair representations, triangular updates) have been shown to be crucial for stability and accuracy in large macromolecular systems.
>
> > 6. Reporting standard deviations
>
> We agree that reporting variance improves comparability. Our experiments already use multiple seeds for baselines (Appendix A), and our sampling procedure is stable across seeds. In the final version, we will add standard deviations for the main GEOM-Drugs metrics.
>
> [1] Chaitanya Joshi et al., All-atom diffusion transformers: Unified generative modelling of molecules and materials, ICML 2025
>
> [2] Ross Irwin et al., Semlaflow – efficient 3d molecular generation with latent attention and equivariant flow matching, ICLR 2025
>
> [3] Tuan Le et al., Navigating the design space of equivariant diffusion-based generative models for de novo 3d molecule generation, ICLR 2024
>
> [4] Ian Dunn et al., Mixed continuous and categorical flow matching for 3d de novo molecule generation, arXiv.2508.12629,  2024

---

### Review · Reviewer_Fj87 · 2025-12-03

**Summary Of Contributions:**

This paper presents TABASCO, a simplified Transformer-based model for 3D molecular generation that avoids explicit $SE(3)$-equivariant layers and relies instead on standard attention mechanisms combined with data augmentation. The authors show that such a non-equivariant architecture can achieve competitive physical plausibility while offering a substantial improvement in sampling speed. The work also provides an extensive set of ablation studies that probe the roles of positional encodings, approximate equivariance, stochasticity, and architectural choices. In addition, the paper proposes a distance-bounds guidance strategy that aims to improve molecular geometry without invoking expensive force-field evaluations.

Key Strengths:
The model demonstrates very large speed gains that make high-throughput 3D molecule generation more practical. TABASCO achieves strong PoseBusters performance, particularly when combined with physical guidance. The breadth of ablation studies is noteworthy and reflects careful scientific investigation of architectural components. The overall design is easy to implement and avoids the complexity typically associated with equivariant neural networks.

Key Weaknesses:
The comparison with the most relevant non-equivariant baseline (ADiT) is not scientifically controlled, since the evaluation relies on pre-generated molecules whose conditions are undocumented. Several claims regarding applicability to structure-based or pharmacophore-conditioned drug design are not supported by any experimental evidence. The model shows limited scaling behavior, as increasing the number of parameters yields only small performance gains. The proposed guidance technique improves metrics but introduces substantial computational overhead and leads to higher strain energies. Some of the mechanistic interpretations, such as the role of positional encodings in scaffold construction or the emergence of approximate equivariance, are largely speculative and not empirically validated.

**Audience:**

Yes

**Audience Explanation:**

While non-equivariant Transformer approaches have been explored in the protein domain, some readers of TMLR may still find the small-molecule results presented in the TABASCO paper relevant and informative.

**Claims And Evidence:**

No

**Claims Explanation:**

1. The paper repeatedly claims that TABASCO is suited to structure- and pharmacophore-based drug design. However, the submission contains no experiments involving conditional generation, receptor-aware design, pharmacophore constraints, docking, affinity evaluation, or any other downstream task. All experiments are limited to unconditional 3D molecule generation. This makes the claim scientifically unsupported.

2. The comparison with the most relevant non-equivariant baseline (ADiT) is not scientifically controlled. ADiT is evaluated using pre-generated samples produced under undocumented sampling conditions, whereas all other baselines are evaluated via re-runs. As a result, the evidence for TABASCO's relative speed and physical plausibility is neither accurate nor convincing.

3. The paper offers several mechanistic interpretations, such as the idea that SMILES positional encodings facilitate early scaffold formation or that approximate equivariance naturally emerges. None of these claims is supported by targeted analyses such as attention visualizations, intermediate-step evaluations, counterfactual orderings, or theoretical arguments. The explanations therefore remain speculative.

4. The narrative suggests that the architecture scales effectively, yet the experiments show that a four-fold increase in parameters yields almost no improvement in PoseBusters scores. No analyses are provided to explain this phenomenon. Consequently, the claim of effective scaling is not supported and is contradicted by the reported results.

**Requested Changes:**

1. Ensure all baseline experiments (e.g., ADiT) are conducted under comparable conditions, and explicitly report limitations and negative results.

2. Either include conditional generation or pharmacophore-guided experiments, or remove unsupported claims from abstract, introduction, and conclusion.

3. Provide attention visualizations, intermediate step analyses, or controlled ablations to demonstrate how positional encodings and approximate $SE(3)$ equivariance affect generation.

---

> ### Author Response · Authors · 2025-12-13
> **Response 1**
>
> We thank the reviewer for the careful reading of our manuscript and the constructive feedback. Below we address each point in turn, and we describe the revisions we make.
>
> > Unsupported claims regarding conditional generation
>
> We intended the current manuscript to focus exclusively on TABASCO’s capabilities in unconditional molecular generation, and deliberately separated these results from any discussion of conditional tasks. The statements regarding conditional or pharmacophore-guided generation in the introduction were meant to be forward-looking, highlighting the architectural extensibility of TABASCO, but we agree that they may read as implying experimental validation. We have therefore revised the manuscript to remove or rephrase these points, ensuring that our contributions and scope are communicated clearly.
>
> > ADiT comparison
>
> We agree that the comparison to ADiT requires additional clarity and rigour. We therefore used the ADiT public repo at https://github.com/facebookresearch/all-atom-diffusion-transformer and the checkpoints provided on HuggingFace to sample and evaluate the model. We observe identical sampling times for 1000 molecules (518.41 s vs. 521.21), however we notice a drop in performance (see tables below with new results). We believe this may be a consequence of the uploaded checkpoint being undertrained, as the HuggingFace file name suggests: https://huggingface.co/chaitjo/all-atom-diffusion-transformer/tree/main/GEOM_only_undertrained. We will be in touch with the authors of the paper to confirm which set of results is most appropriate to report in the final version of the manuscript. To address the reviewer’s comments on the sampling conditions, we can now confirm that the sampling time and parameter count are identical to the original ones reported in our paper.
>
> | Model        | Overall | Connected | Bond Lengths | Bond Angles | No Clashes | Rings Flat | Double Bonds Flat | Internal Energy |
> |--------------|---------|-----------|--------------|-------------|------------|------------|-------------------|-----------------|
> | ADiT | 0.76    | 0.93       | 0.90          | 0.93         | 0.91       | 1.0        | 0.99               | 0.94             |
>
>
> | Method        | # Params. | Validity ↑ | Novelty ↑ | Diversity ↑ | Strain Energy ↓ | Time ↓ (s) |
> |---------------|-----------|------------|-----------|-------------|------------------|------------|
> | ADiT | 150M         | 0.93        | 1.0       | 0.90        |        51.13          |      518.41     |
>
>
> > SMILES Ordering and Approximate Equivariance
>
> We agree with the reviewer that the wording of the claim in Section 3.4 is imprecise, and we will adapt it in the final manuscript. However, we support both claims, each with several experiments.
>
> We provide intermediate-step evaluations and invariant ordering in Section 4.6 to analyze approximate equivariance. We show that the relative equivariance error of the model is less than 2% throughout denoising and significantly lower (0.3% - 0.001%) for large parts of the denoising process. By contrasting the relative equivariance error to that of a model that is trained fully without random rotations, we provide strong evidence that the model exhibits emergent equivariant behavior that is mostly driven by applying random rotations to the data.
>
> For SMILES ordering we analyze the effect of positional encodings in Section 4.5 by again evaluating denoising capabilities at intermediate steps and invariant orderings in Figure 4. Figure 4 left highlights that at intermediate steps the physical plausibility of generated molecules is significantly higher for the first 70% of denoising, when atoms cannot be distinguished by their coordinates. This trend flips for late intermediate steps, when atoms can easily be distinguished by their coordinates. In Figure 4 right we show that full denoising with positional encodings yields significantly more physically valid molecules. We give further evidence in Table 7 and Figure 7 where we show that on trajectories where the point cloud collapses, atoms cannot be distinguished by their coordinates at all, and positional encodings are thus essential to create physically valid molecules (Table 7, Rows 2 and 6).
>
>
> > Scaling the Model
>
> In Section 4.3 we highlight that the only minor performance increase by scaling the model is likely due to the fact that the training dataset GEOM-Drugs itself has only 94% physical validity as measured by PoseBusters. This effectively creates a glass ceiling on performance improvements on the PoseBusters metric, which is already almost saturated by the 15M parameter model at 91%.
>
> We agree that the current formulation could be more precise and will adapt the manuscript to highlight that the model architecture scales efficiently with respect to computational cost (as highlighted by the sampling speed in Table 2).
>
> ---
>
> We are grateful for the reviewer’s feedback and are glad to elaborate further or make any necessary corrections.

---

### Decision · Action_Editor_Ggmz · 2026-01-09

**Recommendation:** Accept with minor revision

**Additional Comments:**

I recommend acceptance subject to the minor revisions highlighted by reviewers:

Authors must address the discrepancy in the ADiT baseline performance. As noted in the discussion, the authors should ensure they are not comparing against an undertrained checkpoint. The final camera-ready version must clarify the source of the baseline results and ensure the comparison is fair, or explicitly discuss why the reproduction yielded different results than the original ADiT publication.

Authors should expand the limitations section to explicitly mention the "scaling plateau" observed (performance not improving significantly beyond 15M parameters) and the current limitations regarding mechanistic understanding of the emergent equivariance, as requested by Reviewer Fj87.

Finally, authors must ensure the manuscript clearly delineates that the current validation is for unconditional generation only, as discussed in the response to Reviewer Hc1X.

**Audience:**

Yes

**Audience Explanation:**

The paper addresses a central topic in AI4Science and machine learnin (molecular generation). By challenging the prevailing assumption that expensive SE(3)-equivariant layers and graph-based message passing are strictly necessary for physical plausibility, the work offers a "simplification" perspective that is highly relevant to researchers looking to scale generative models. The findings regarding the efficacy of standard Transformers and flow matching in this domain will be of significant interest to the TMLR audience.

**Claims And Evidence:**

Yes

**Claims Explanation:**

All reviewers have reached a consensus that the primary claims of the paper, like the computational efficiency and physical validity of the proposed TABASCO model are supported by the provided experiments. Reviewer Fj87 initially raised concerns regarding the rigor of the ADiT baseline comparison, authors have acknowledged this and the reviewer has subsequently updated their recommendation noting that the core contributions remain valid and the baseline issues are resolvable through revision. The empirical evidence that a non-equivariant architecture can achieve state-of-the-art performance on specific benchmarks is convincing.

---

> ### Author Response · Authors · 2026-02-10
>
> Thank you for the decision and the constructive feedback from all reviewers.
> We have revised the manuscript to address the requested changes:
>
> 1. We clarified the wording throughout to emphasize that the model is for unconditional generation only.
> 2. We added a note in Appendix A addressing the discrepancy in ADiT baseline performance. We chose to keep the higher values based on the provided molecules since the provided ADiT model checkpoints are from an independent reproduction with lower values and we believe this gives a fairer comparison.
> 3. We extended the Discussion and Limitations sections to cover the scaling plateau and limited mechanistic understanding of emergent equivariance, as raised during review.
>
> Please let us know if anything else needs attention.